# Insights into Alkaline Phosphatase Anti-Inflammatory Mechanisms

**DOI:** 10.3390/biomedicines12112502

**Published:** 2024-11-01

**Authors:** Larissa Balabanova, Georgii Bondarev, Aleksandra Seitkalieva, Oksana Son, Liudmila Tekutyeva

**Affiliations:** 1G.B. Elyakov Pacific Institute of Bioorganic Chemistry, Far Eastern Branch, Russian Academy of Sciences, Prospect 100-Letya Vladivostoka 152, 690022 Vladivostok, Russia; sasha0788@inbox.ru; 2Youth Research Laboratory of Recombinant DNA Technologies, Advanced Engineering School, Institute of Biotechnology, Bioengineering and Food Systems, Far Eastern Federal University, 10 Ajax Bay, Russky Island, 690922 Vladivostok, Russia; bondarevgeorgii22@gmail.com (G.B.); oksana_son@bk.ru (O.S.); tekuteva.la@dvfu.ru (L.T.)

**Keywords:** alkaline phosphatase, LPS-induced inflammation, TLR4-NF-kB signaling pathways, immune metabolism, pro-inflammatory metabolism, catabolic phenotype, mitochondrial biogenesis, oxidative phosphorylation, endocytosis, autophagy

## Abstract

Background: The endogenous ecto-enzyme and exogenously administered alkaline phosphatase (ALP) have been evidenced to significantly attenuate inflammatory conditions, including Toll-like receptor 4 (TLR4)-related signaling and cytokine overexpression, barrier tissue dysfunction and oxidative stress, and metabolic syndrome and insulin resistance, in experimental models of colitis, liver failure, and renal and cardiac ischemia-reperfusion injury. This suggests multiple mechanisms of ALP anti-inflammatory action that remain to be fully elucidated. Methods: Recent studies have contributed to a deeper comprehension of the role played by ALP in immune metabolism. This review outlines the established effects of ALP on lipopolysaccharide (LPS)-induced inflammation, including the neutralization of LPS and the modulation of purinergic signaling. Results: The additional mechanisms of anti-inflammatory activity of ALP observed in different pathologies are proposed. Conclusions: The anti-inflammatory pathways of ALP may include a scavenger receptor (CD36)-mediated activation of β-oxidation and oxidative phosphorylation, caveolin-dependent endocytosis, and selective autophagy-dependent degradation.

## 1. Introduction

While the underlying mechanisms of inflammation are well-established, the immune response involving the alkaline phosphatase (ALP) pathway remains poorly understood, particularly with regard to the development of anti-inflammatory and anti-cancer drug targets [1,2,3,4,5,6,7,8,9,10,11,12].

It is established that inflammatory processes result in the dysfunction of epithelial tissues, thereby facilitating the entry of resident and pathogenic microflora antigens, including bacterial endotoxic lipopolysaccharide (LPS), into the bloodstream [2,3,8,13,14,15,16,17]. The presence of bacterial antigens has been observed to contribute to the exacerbation of local inflammatory processes in injured tissues, thereby amplifying the disruptive changes that occur as a result of the initial injury. Once in the bloodstream, these substances cause systemic inflammation and damage to vital organs, which in severe cases can lead to sepsis and death [13,16]. Tissue-specific intestinal alkaline phosphatase (IAP) has the capacity to recognize and dephosphorylate bacterial LPS, reducing its antigenic properties and resulting in a reduction in local inflammation, tissue permeability, and systemic immunotropic effects [1,2,6,8,10,11,13,15,18,19]. It thus follows that IAPs have been classified as part of the ancient innate immune system of multicellular organisms, which have a protective effect against inflammation of their tissues during microbial colonization [1,2,13,14,19]. In general, IAPs have the potential to treat acute and chronic pathologies associated with aberrant activation of LPS-specific Toll-like receptor 4 (TLR4)-mediated pathways in epithelial and immune cells (neutrophils and macrophages) [1,2,6,8,10,11,13,15,18,19].

Furthermore, IAP can either directly or indirectly neutralize other pathogen- or damage-associated molecular patterns (PAMP or DAMP) produced in response to LPS and non-LPS-mediated inflammation. This includes excess purinergic signaling nucleotides (adenosine triphosphate (ATP), adenosine diphosphate (ADP), and adenosine monophosphate (AMP)), nicotinamide adenine dinucleotide phosphate (NADPH) cofactors and precursors, and mitochondrial components released from damaged cells by reactive oxygen species (ROS) [1,4,5,6,10,18,20,21,22,23,24]. The primary sources of ROS implicated in chronic health conditions such as inflammatory bowel disease, metabolic syndrome, chronic kidney disease, and ischemia-reperfusion injury are NADPH oxidases (NOX) and mitochondria, which regulate the biochemical processes of respiration and energy production in immune and tissue cells [17,18,21,23,25,26,27,28,29,30,31,32].

Peripheral blood ALPs, predominantly the tissue non-specific alkaline phosphatase (TNAP) isozyme, have been observed to dephosphorylate a range of inflammatory factors, including antigens, free nucleic acids, proteins, lipids, and pyrophosphate (PP_i_), which are absorbed into the blood at the site of inflammation [1,3,5,7,24,33,34]. Furthermore, the administration of exogenous ALP has been demonstrated to significantly attenuate inflammation in murine models of colitis, acute-on-chronic liver failure, and renal and cardiac ischemia-reperfusion injury and to provide protection in human proximal tubule epithelial cells [3,4,6,10,11,35,36,37,38,39]. Recombinant IAPs have been shown to mitigate oxidative stress, alter intestinal barrier protein expression, modulate gut microbiota, and ameliorate insulin resistance, as well as prevent metabolic syndrome in experimental obese mice with colitis. These findings suggest that the anti-inflammatory action of ALP is multifaceted [1,6,11,12,35,39].

Given that anti-inflammatory activity has been observed in the absence of LPS in leukocytes and in the A2A adenosine receptor knockout model in injured kidney tissue, it can be postulated that the anti-inflammatory mechanisms of both the exogenous and endogenous ALP ectoenzyme are not confined to the detoxification of LPS and the production of anti-inflammatory adenosine from ATP, ADP, and AMP [4,35,38,40]. It is noteworthy that ALP-deficient progenitor cells displayed mitochondrial hyperfunction, which resulted in impaired proliferation and differentiation [41].

Recent studies have provided new insights into the mechanistic role of ALP in immune metabolism. The anti-inflammatory ALP pathways, which have yet to be determined, may include mitochondrial biogenesis through the activation of β-oxidation and oxidative phosphorylation, caveolin-dependent endocytosis, and selective autophagy-dependent degradation [7,11,12,17,40,41,42,43,44,45].

## 2. Alkaline Phosphatase Inhibit LPS-TLR4 Binding and TLR4-Mediated NF-κB Signaling

The precise role of the LPS-induced TLR4-mediated nuclear factor kappa-light-chain-enhancer of activated B cells (NF-κB) pathways and ALP isoenzymes in the inflammatory response and tissue barrier dysfunction remains unclear [10,18,46]. Tissue barrier dysfunction can be observed in the context of various factors, including inflammatory processes, dysbiosis, intoxication, diet, stress, and genetic and environmental factors such as mutations or low levels of ALP activity [2,3,8,13,21,47]. Dysfunction of the tissue barrier allows antigens, particularly the bacterial endotoxin LPS, to enter the bloodstream. The largest source of resident microflora or pathogenic LPS is the gut [1,2,8,11,13]. LPS represents a significant category of pathogen-associated molecular pattern (PAMP) that is identified by the TLR4/myeloid differentiation factor-2(MD2) complex, which constitutes a component of the innate immune system [13,14,15]. LPS activates transmembrane NADPH oxidases (NOX) via the transcription factor NF-kB, a key mediator of the inflammatory response, by targeting TLR4 on innate immune cells (neutrophils and macrophages) from the lamina propria and enterocytes [2,6,18,46]. For example, treatment of macrophages with LPS has been shown to increase the classical immune response through an oxidative burst via the isoenzyme NADPH oxidase 2 (NOX2) [17]. The NOX2 isotype enhances the production of reactive oxygen species (ROS), including the superoxide anion (O^2−^), nitric oxide (NO), peroxynitrite (ONOO^−^), and hydrogen peroxide (H_2_O_2_), as well as hypochlorous acid (HOCl), in neutrophils and macrophages (Figure 1). This results in the release of pro-inflammatory cytokines, including IL-1β, IL-6, and TNF-α, which facilitate the recruitment of leukocytes to the ROS-damaged tissue [6,18,27,48]. ROS play a role in the function of leukocytes; however, the presence of excess oxidants can result in tissue damage and oxidative stress [49,50,51]. Furthermore, the isoenzyme NADPH oxidase 1 (NOX1) represents a principal source of LPS-induced ROS in non-phagocytic intestinal epithelial cells, thereby exerting a pivotal influence on LPS-induced intestinal hyperpermeability through the activation of matrix metalloproteinase-9 (MMP9) and the promotion of inflammatory processes [21] (Figure 1).

The NF-κB pathways not only directly increase the production of pro-inflammatory molecules but also contribute to cell proliferation, apoptosis, morphogenesis, and differentiation [46]. As a consequence, the inhibition of TLR4-mediated NF-κB signaling in LPS-induced macrophages was associated with a reduction in the expression of cyclooxygenase-2 (COX-2), nitric oxide synthase (iNOS), tumor necrosis factor alpha (TNF-α), interleukin 1 beta (IL-1β), and IL-6 production, and of phosphorylation of serine/threonine-specific protein kinases (AKT) and mitogen-activated protein kinases (MAPKs) [27,52]. Conversely, AMP-activated protein kinase (AMPK) in macrophages has been shown to restrict the progression of inflammation in experimental colitis by suppressing NOX2 expression, ROS generation, and pro-inflammatory cytokine secretion [25,53]. Nevertheless, recent findings from AMPKβ1-myeloid-deficient mice indicate that the inhibition of NOX2 may potentially reduce inflammation independently of the AMPK pathway, which is an antagonist of the NF-kB pathway [25,27].

Intestinal ALP isozyme IAP has been reported to constitute part of the innate immune system and to exert a protective effect during inflammatory processes [1,2,13,14,19]. Tissue-specific IAP molecules are expressed by enterocytes into the membrane and are also released from brush border microvilli into the intestinal lumen in the form of vesicles, subsequently entering the bloodstream [54]. IAP dephosphorylates bacterial LPS and thereby reduces its antigenic properties, which in turn leads to a reduction in local inflammation, intestinal permeability, and systemic immunotropic effects. Consequently, IAP could potentially be employed in the treatment of acute and chronic pathologies involving aberrant TLR4 activation [1,2,13,14,19] (Figure 1). Despite the recent discovery that ALP specificity for the different types of enteric LPS is limited by the degree of lipid A acylation, removal of one of its two lateral phosphate groups by the enzyme resulted in a the 100-fold reduction in LPS toxicity [13,15]. The dephosphorylated LPS retained the ability to bind to TLR4 but acted predominantly as an antagonist. The reduction in LPS toxicity resulted in the inhibition of downstream intracellular signaling, which led to the suppression of the activated NF-κB and subsequent cellular response, including the release of pro-inflammatory cytokines, chemokines, adhesion molecules, and vasodilation, as well as the recruitment of immune cells and plasma proteins to the site of infection or tissue injury [2,6,13,18,46,48] (Figure 1).

It is noteworthy that the commercially available ALP preparations, including calf and human IAPs, have minimal in vitro activity against fully acylated enteric LPS chemotypes. In order for them to be dephosphorylated, their partial or complete deacylation is required [15]. This may explain the partial effect of exogenous ALP in numerous trials for the treatment of multiple chronic metabolic endotoxemia or sepsis, where LPS-induced inflammation is a suspected component of the underlying pathology. However, the results of the use of ALP as a treatment are further complicated by the heterogeneity of clinical data and individual immune system characteristics, as well as the time dependency if ALP is administered too late after the inflammation has developed [10]. In any case, the anti-inflammatory pathways of ALP appear to be considerably more complex and extensive than was previously thought, as evidenced by the literature [4,6,10,41].

For example, in vitro studies have demonstrated that IAP can inhibit the release of TNF-α and IL-6 by freshly extracted human leukocytes in the absence of LPS, as well as attenuate oxidative stress, alter intestinal barrier protein expression, modulate the gut microbiota, and alleviate insulin resistance in experimental obese mice with colitis, suggesting a multifaceted mechanism of anti-inflammatory action of ALP [3,4,6,10,11,12,35,36,37,38,39]. Furthermore, IAP is understood to play a role in maintaining duodenal surface pH, facilitating the absorption of long-chain fatty acids, and neutralizing PAMP and DAMP in the intestinal lumen, including flagellins, DNA, RNA, nucleotides, and cofactors [2,6,7,8,10,11] (Table A1).

## 3. Alkaline Phosphatase Modulate Purinergic Signaling

ALP isoenzymes can either directly or indirectly neutralize either bacterial PAMP or host DAMP produced during immune or inflammatory responses to various stimuli (canonical NF-κB and alternative pathways), including ligands of various cytokine receptors, pattern recognition receptors (PRRs), members of the TNF receptor (TNFR) superfamily, and T cell (TCR) and B cell receptors [46]. Moreover, neutrophil extracellular traps (NETs) are regarded as a source of autoantigens that contribute to chronic inflammation and autoimmune disorders in the absence of DNAse and ALP-mediated degradation of cytotoxic phosphorylated NET-bound cell-free DNA and granule proteins [55,56].

It can be reasonably deduced that the observed anti-inflammatory effects of ALP are most likely due to the effective dephosphorylation of intercellular signaling nucleotides (Table A1) rather than detoxification of LPS, in particular the production of adenosine (Ado) from purines, which has been shown to possess anti-inflammatory properties [4,5,6,20,22,35]. Signaling adenosine is the end product of the non-specific ALP-mediated enzymatic degradation of extracellular ATP, ADP, and AMP, which are the substrates for the specific ecto-enzymes CD39, CD73, and ectonucleoside triphosphate diphosphohydrolase 1 (eNPP1) that regulate the expression of purinergic signaling receptors [20] (Figure 1). Adenosine has been shown to downregulate ATP-binding P2-type purinergic receptors on immune cells and to exert anti-inflammatory effects by binding to A2A receptors that inhibit NF-κB, TNF, and phosphatidylinositol 3-kinase/protein kinase B (PI3K-Akt) pathways [6,22] (Figure 1). Consequently, the concentration of extracellular ATP and ADP was found to increase following incubation with LPS in the HT-29 model epithelial cells [6]. Nevertheless, pre-incubation with an animal IAP was observed to abolish this increase, resulting in the release of Ado and its involvement in the anti-inflammatory A2A-mediated signaling pathways. However, when HT-29 cells were treated with adenosine, there was a marked increase in the expression of transcripts related to the nucleotide-binding oligomerization domain (NOD)-like receptors, the Janus kinase/signal transducers and activators of transcription (Jak-STAT), chemokines, PI3K-Akt, and TNF pathways. In contrast, transcripts of Toll-like receptors and TLR4-linked NF-kB pathways were predominant in the cells incubated with an IAP + LPS mixture [6].

The anti-inflammatory effect of ALP in LPS- and non-LPS-induced inflammation does not appear to be limited by LPS detoxification and adenosine production from released purines [4,12,38,41]. For example, Gao et al. [6] showed that the anti-inflammatory activity of IAP is associated with the dephosphorylation of other nucleotides (Table A1), which act synergistically to enhance the action of adenosine. The combination of guanosine, cytidine, and thymine with adenosine demonstrated enhanced efficacy in the inhibition of TNF-a, IL-6, and cell adhesion and migration [6]. In addition to ALP’s ability to cleave nucleotides, PP_i_, vitamin B6, phosphoethanolamine, phosphocholine, and phosphopeptides [24,34], ALP has the capacity to dephosphorylate the key transmembrane regulators of pro-inflammatory pathways, including phosphoinositides (PIs), transmembrane receptors, kinases, and protein phosphatases (Figure 1). This prevents their activation and their exertion of anabolic effects on cells, thereby reducing the risk of mitochondrial DNA mutation. This, in turn, has the potential to enhance bioenergetic function and reduce glucose dependence [57,58].

## 4. Alkaline Phosphatase Modulate Pro-Inflammatory Metabolism

The innate immune response to bacterial infection is regulated by a metabolic switch from oxidative phosphorylation to glycolysis in macrophages exposed to LPS [17,18]. In particular, glycolysis and the pentose phosphate pathway (PPP) are essential for the promotion of LPS-induced pro-inflammatory metabolism in macrophages [17,18]. Consequently, the repression biochemical markers linked to pro-inflammatory metabolism in macrophages (M1) enables monocytes to differentiate into macrophages belonging to the resolution phase of inflammation (M2) [18,52] (Figure 1). While M1 macrophages and phagocytic neutrophils rely on NOX2 activation, which increases glucose uptake transporter (GLUT4) expression and glycolysis-derived energy production, to drive their migration and ROS-mediated bacterial killing, anti-inflammatory M2 macrophages, as resting lymphocytes, rely mainly on mitochondrial oxidative phosphorylation (oxphos) for energy accumulation [18] (Figure 1). Indeed, chronic inflammation is associated with increased mitochondrial superoxide, oxidative damage, and innate immune senescence without resolution of the inflammatory event [13,18,32,36]. The anti-inflammatory activity of ALP against LPS may be associated with catabolic processes, which have been linked to the reduced expression of metabolic pathway genes and an inhibitory effect on the progression and chronicity of inflammation. This is supported by evidence from studies [2,12,40]. To illustrate this, the activity of hepatic LPS was diminished by a recombinant ALP that prevented multiple organ injury by reducing TLR4 expression in acute-on-chronic (but not acute) liver failure [36]. Alkaline phosphatase plays a significant role in reducing intercellular permeability, presumably by affecting the expression of the metalloprotease MMP9 and its inducing proteins, such as the non-phagocytic isoform of the superoxide-generating NOX1 [10,25] (Figure 1).

It has been proposed that the metabolic control of a cellular phenotype switch from metabolically active to catabolic pathways associated with NOX inhibition is mediated by redox reactions and the activity of NAD^+^ salvage and de novo NAD^+^ synthesis pathways. These include the proteins CD38, CD39, CD73, NAMPT, PARPs, sirtuins SIRT1-7, etc. [7,18,20,59]. In the presence of active NAD^+^-dependent SIRT1, the inhibition of inflammation and glycolytic metabolism was observed, while the promotion of mitochondrial biogenesis and fatty acid oxidation was noted [18,59]. Moreover, it is established that NAD^+^-mediated energy signaling plays a role in the development of metabolic syndrome, cancer, and ageing [60]. It seems probable that the catalytically non-specific ALP, in conjunction with the specific membrane-bound ATP hydrolase CD39 and the AMP-hydrolyzing ecto-5′-nucleotidase CD73, serves to rapidly degrade purines as well as NAD(P)H released from damaged host and microbial cells, thereby maintaining immunoregulatory function via purinergic and NAD^+^ signaling [5,20] (Figure 1).

In addition to degradation of key precursors and cofactors for NADPH-dependent NOX in inflammatory cells, mitochondrial respiration and energy (ATP) production via glycolysis and PPP should be reduced in order to switch from the metabolically pro-inflammatory to the catabolic phenotype [7,18,41].

It is hypothesized that ALP functions to remove extracellular phosphates (P) from a transmembrane B class scavenger receptor CD36 (SR-B2), a translocase that responds to the balance of external fatty acid levels (Figure 1). This results in the activation of their uptake for β-oxidation and the subsequent induction of mitochondrial biogenesis and the switch to oxidative phosphorylation in macrophages (from M1 to M2) by the inhibition of NF-κB-mediated pro-inflammatory expression [7,18,27,44,45,61] (Figure 1). In an experimental model of colitis in obese mice, Wojcik-Grzybek et al. [11] observed that the administration of IAP in conjunction with moderate physical activity was associated with a reduction in adipose tissue accumulation, improved insulin sensitivity, and a lower incidence of metabolic syndrome. Additionally, this treatment regimen was associated with a mitigation of colonic inflammation, as evidenced by a modulation of the gut microbiota, proinflammatory cytokines, and oxidative DNA damage in the colonic mucosa. In contrast, IAP knockout mice were observed to gain weight when fed a high-fat diet (HFD) [39].

However, it has been demonstrated that upon stimulation by insulin or muscle contraction, CD36 translocation to lipid rafts of the cell membrane and fatty acid uptake may be associated with the activation of either the insulin-based PI3K-Akt pathway or the CD36-controlled negative feedback AMPK pathway [45]. The discrepancy between the functions of CD36 may be attributed to the concentration of insulin and the differentiation between the vesicular trafficking pathway and the subcellular trafficking of CD36 and GLUT4 [45]. The insulin-induced translocation of CD36 and subsequent uptake of fatty acids are associated with the concomitant translocation of GLUT4 and glucose uptake. Conversely, AMPK activation-induced CD36 translocation and fatty acid uptake are associated with free fatty acid oxidation [62,63]. The vesicle-associated membrane protein (VAMP) VAMP4 is responsible for the translocation of CD36 between the parent endosomal compartment and a hypothetical CD36-specific intermediate endosomal compartment. In contrast, VAMP5 and VAMP7 are involved in the homing of GLUT4 to intracellular compartments [45]. Furthermore, CD36 maturation and trafficking are regulated by post-translational modifications, including phosphorylation, ubiquitination, glycosylation, and palmitoylation, which are responsive to tissue- and environment-specific stimuli. For these reasons, it may be posited that CD36 deficiency may prove an effective means of ameliorating diabetic cardiomyopathy and atherosclerosis, whereas CD36 overexpression may serve to reverse ischemia-reperfusion injury. For example, CD36-deficient epithelial cells that were fed an HFD exhibited enhanced glucose tolerance and insulin sensitivity [64], while skeletal muscle-specific CD36 deletion resulted in a reduction in the expression of genes involved in insulin signaling and glucose metabolism [65].

## 5. Alkaline Phosphatase-Phosphate Complexes Trigger Autophagy and Endocytosis

It has recently been demonstrated that the anti-inflammatory activity of ALP is dependent on autophagy-mediated degradation. The impact of IAP on TLR4 signaling and lysosomal gene expression in relation to autophagy was substantiated in enterocytes and LPS-induced macrophages [12,40]. The anti-inflammatory effect of IAP in an autophagy-dependent manner was corroborated by the inhibition of IL-1β mRNA expression, which was achieved by interfering with LPS-mediated activation of NF-κB (RelA/p65 phosphorylation). LPS was observed to induce increased phosphorylation of the p65 subunit, which activates the NF-κB inflammatory pathway. However, IAP was demonstrated to significantly inhibit this process [12]. Nevertheless, it is as yet unclear whether IAP acts directly or indirectly, downstream of TLR4 activation or upstream of NF-κB signaling.

Remarkably, ALP-deficient progenitor cells exhibited mitochondrial hyperfunction and increased ATP production, which resulted in disturbances to their proliferation and differentiation [41]. It is therefore hypothesized that ALPs affect mitochondrial function by generating high levels of inorganic phosphate (P_i_), which prevents mitochondrial membrane depolarization and thus mitochondrial participation in oxidative stress signaling pathways [21,41,66]. Similarly, the same PhoA family ALPs of marine bacteria and microalgae have been proposed to deliver P_i_ to the periplasmic space to maintain redox and cell cycle balance. This is based on their single-operon location together with the electron transport chain proteins within the chromosome [67]. Given that the ecto-enzymes ALP have been observed within progenitor cells in close proximity to the nucleus and mitochondria [41], it is plausible that de novo synthesized ALP may be embedded in an early phagophore membrane derived from the endoplasmic reticulum (ER) and attached to mitochondria and/or the lysosomal membrane in close proximity to mitochondria at a specific PI3P-enriched site (omegasome) [68,69,70,71] (Figure 1).

On the other hand, an interaction between the overexpressed glycosylphosphatidylinositol (GPI)-bound ALP on the plasma membrane and their substrates has been shown to induce caveolae-mediated endocytosis, which shares a common endpoint with autophagy at the lysosome where its ‘cargo’ is degraded [70,72,73]. Thus, the cellular uptake of complex organic phosphates (phosphopeptides) was found to be more dependent on interactions between the alkaline phosphatase TNAP and the enzyme triggers (substrate assembly-induced aggregation) than on binding between receptors and their ligand structures. TNAP catalyzed the further dephosphorylation of the multimeric phosphopeptides in endosomes [73]. Furthermore, it has been shown that the enzymes IAP from aggregates in the presence of inorganic phosphate, yet this does not result in a reduction in their catalytic activity [74].

The mechanism elucidated by He et al. [73] is likely to be applicable to the endocytosis and endosomal escape of other multimolecular substrates of ALP, including LPS at concentrations approaching the critical micelle concentration (CMC). At high levels of TLR4 expression, the LPS bound to TLR4 on the plasma membrane can bind an excess of endogenous or exogenous ALP (ALP or eALP) due to their mutual affinity, thereby triggering and/or accelerating endocytosis-based LPS escape [2,12,42,43] (Figure 1). TLR4 is known to be selected as a cargo for inflammatory endocytosis entirely through extracellular interactions, and the selective autophagy-dependent degradation of TLR4 is related to an ancient mechanism for endotoxin clearance and immune regulation [42,43]. In addition, Groza et al. [75] have demonstrated that lipid-binding-mediated endocytosis occurs with a high degree of reliability once a threshold of adhesion energy of the multivalent globular binders is surpassed, thereby allowing for membrane deformation. Previously, it was demonstrated that the lipid raft-associated isoform IAP is rapidly and selectively (although clathrin-dependently) endocytosed from the morphologically altered regions of brush border membranes into intracellular compartments during the physiological process of fat absorption. This limits the rate of dietary lipid uptake by enterocytes [76].

In myocardial tissue, the transmembrane fatty acid translocase CD36, which is localized in lipid rafts of the plasma membrane and interacts with caveolins, has been shown to be activated by ALP-mediated dephosphorylation of two extracellular phosphate groups of CD36, which may optimize fatty acid uptake through caveoline-dependent endocytosis [4,44,45,61]. In this manner, ALP may facilitate β-oxidation and the subsequent AMPK pathway through CD36, which induces mitochondrial biogenesis and a transition to oxidative phosphorylation in M2 macrophages by inhibiting NF-κB-mediated pro-inflammatory expression [7,18,27,44,62] (Figure 1). CD36 is not only solely present on plasma membranes and endosomes; it is also distributed on mitochondria, presumably alongside ALP to fulfil the same function [45].

Furthermore, it is possible that extracellular vesicular and intercellular non-vesicular transfer of endogenous ALP containing C-terminal GPI may occur, mediated by homo- or heteromeric aggregates and lipoprotein- or lipid-containing micelle-like complexes [77]. This could potentially induce switching phenotypes upon proximity or contact with somatic cells/tissues. It is noteworthy that, following the induced lipolytic release from plasma membranes, these endogenous ALP displace serum GPI-binding ALP and assemble into aggregates or micelle-like complexes [77]. It seems probable that such ALP-containing aggregates will bind LPS due to substrate affinity, as well as adhere to the inflammation-induced phosphoinositide (PI)-specific membrane sites that are predominant for PI3K/Akt signaling. The PI3K/Akt signaling pathway is negatively regulated by the recruitment of the phosphatase/phospholipase PTEN to the membrane [58,68,71] (Figure 1). PTEN itself functions as both an activator and a substrate of chaperone-mediated autophagy, as well as a substrate for non-specific ALP-mediated cleavage of its four C-terminal phosphates. The dephosphorylation of the protein by ALP can allosterically activate PTEN, thereby inhibiting the PI3K/Akt/Mammalian target of rapamycin(mTOR) pathway, mitochondrial function, and the pro-inflammatory response of innate immune cells [57,58,78] (Figure 1).

The turnover of phosphoinositides (PIs) in response to alterations in energy metabolism also plays a role in the regulation of NOX activity [49,71,79] (Figure 1). The activation of NOX2 by the assembly of its subunits at the plasma or phagosomal membrane is dependent on the availability of NADPH and the ratio guanosine di-/triphosphate (GDP/GTP), as well as on the interaction of Rho-associated protein kinase 2 (ROCK2) with the membrane subunit p22phox. This enables the phosphorylation of the translocated cytosolic regulatory subunit p47phox, which in turn increases ROS production and maintains a positive feedback loop for phagocyte migration and ROS production in epithelial cells [49,50,51,78,79,80]. The consumption of oxygen and protons by NOX2 for ROS production alkalizes the phagosomal environment, thereby facilitating antigen processing and increasing cross-presentation, as well as favoring ALP activity [81]. During the process of phagosomal membrane internalization, the phosphatidylinositol 3-phosphate (PI3P)-specific site is generated through the dephosphorylation of PI(3,4)P2 and the phosphorylation of phosphatidylinositol by class III PI3K. These products are responsible for maintaining the cytosolic NOX2 subunits at the plasma membrane, presumably via PI binding of the PX domain of p47phox. Therefore, the various PIs and their associated phosphatases regulate protein trafficking to the phagosome and autophagosome, as well as to the early and recycling endosome (Figure 1).

It is challenging to assess the reactivity of ALP towards the phosphate groups of membrane-bound PIs and transmembrane proteins, particularly those that are phosphorylated on the cytosolic side, in vivo. However, due to the phosphomonoesterase polyspecificity of ALP towards signal transduction components, including immune response signals, it can be postulated that ALP may act as a scavenger enzyme, inducing catabolic dephosphorylation to facilitate the completion of destructive processes that are necessary for cell survival. This does not contravene the common feature of all these processes, namely the utilization of the fundamental physical and chemical laws of nature, including “protein-protein” interaction, the affinity of enzymes for substrates, and the invagination of membranes under multivalent “protein-ligand” complexes [73,75].

## 6. Conclusions

The anti-inflammatory mechanisms of ALP are expected to be universally applicable to both LPS-induced and non-LPS-induced inflammatory processes. This provides a potential explanation for the attenuating effect of ALP in a number of unrelated inflammatory pathologies. ALP-mediated dephosphorylation of LPS and the purine nucleotides ATP, ADP, and AMP is a crucial mechanism for preventing pro-inflammatory signaling, as demonstrated by a large number of in vitro and in vivo experiments. However, the function of ALP pathways in inflammatory processes that do not involve LPS induction or the presence of adenosine receptors remains uncertain. The resolution of inflammatory processes and the subsequent repair of damaged tissue depend on the extent to which the products of pro-inflammatory metabolism are eliminated. This depends on a metabolic transition from an active energy-consuming state, mediated by glycolysis and the pentose phosphate pathway (PPP), to an energy-storing and catabolic phenotype of cells, mediated by oxidative phosphorylation. It seems reasonable to postulate that such a universal switch could be the exogenous and endogenous non-specific phosphomonoesterase ALP, capable of binding a variety of complex phosphates on extracellular and intracellular membrane surfaces (LPS, CD36, NOX, PTEN, PIs, etc.). ALP-bound pro-inflammatory phosphoproteins and phospholipids are internalized and degraded via endocytosis and autophagy, resulting in the silencing of pro-inflammatory signaling pathways and a shift of cells from a pro-inflammatory, metabolically active state to a catabolic phenotype.

It follows that the non-specific phosphomonoesterase ALP, as part of the innate immune system, is involved in the anti-inflammatory adenosine production, LPS detoxification, and CD36-mediated activation of β-oxidation and may act as a scavenger protein that binds and catabolically degrades complex free and membrane-bound signaling phosphoproteins and phospholipids. This is achieved by reducing energy production from mitochondrial glycolysis. Therefore, the diverse effects of non-specific ALP may be relevant to mitochondria-targeting drugs for the control of obesity, diabetes, metabolic syndrome, cancer, and a range of systemic and acute inflammatory conditions.

## Figures and Tables

**Figure 1 biomedicines-12-02502-f001:**
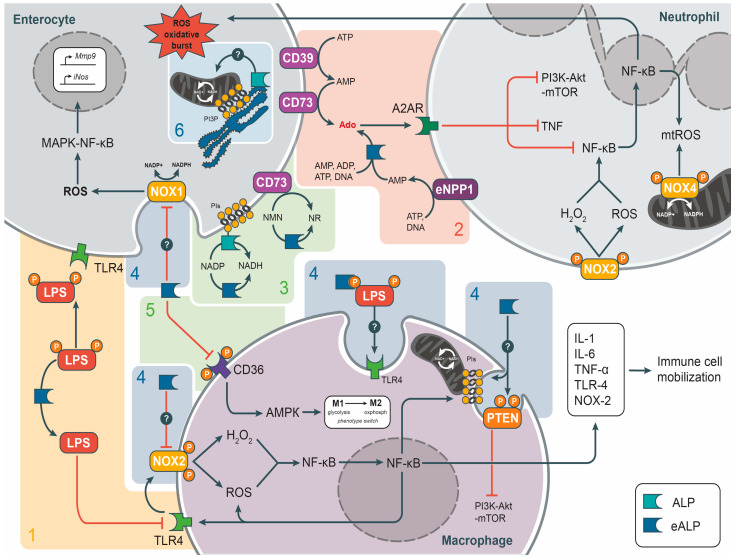
Putative anti-inflammatory mechanisms of non-specific phosphomonoesterase, cellular (ALP: GPI-bound), and extracellular (eALP: exogenous or free endogenous) alkaline phosphatase on innate immune and epithelial cells: (1) Inhibition of LPS-TLR4 binding and TLR-4-NF-kB-induced pathways (IL-1, IL-2, TNF-α, TLR4, NOX2, etc., ROS production (extracellular ROS and mitochondrial mtROS) associated with immune cell recruitment (highlighted by a yellow block in the lower left corner); (2) Dephosphorylation of the intercellular signaling purine nucleotides AMP, ADP and ATP (and pyrimidines), together with the ecto-enzymes CD39, CD73 and eNPP1, to the anti-inflammatory adenosine (Ado) and phosphate ions PO_4_^2−^ (P), allowing Ado to bind to adenosine receptors (A2AR) and inhibit the pro-inflammatory pathways NF-kB, PI3K-Akt-mTOR and TNF-α (illustrated for neutrophil in the upper right corner) (highlighted by a rose block); (3) Dephosphorylation of nicotinamide mononucleotide (NMN) to nicotinamide riboside (NR) and NAD(P)H to NADH for de novo NAD^+^ synthesis and salvage pathways (highlighted by a green block); (4) Putative (?) induction and/or acceleration of the internalization of membrane-bound phosphorylated biomolecules at phosphoinositide(PI)-specific sites, which are recruited by inflammatory signals for caveolae-mediated endocytosis of LPS-bound receptors TLR4 (shown for macrophage, center), NOX1 (shown for enterocyte, top left), NOX2, PTEN (shown for macrophage, center) and other proinflammatory multimolecular factors, due to the ability of ALP (and eALP) to form “enzyme-substrate” complexes or aggregates with LPS, PIs, membrane phospholipids and transmembrane phosphoproteins (highlighted by a blue blocks); (5) dephosphorylation to activate the fatty acid translocase CD36 (receptor SR-B2) to optimize fatty acid uptake, including by caveolae-dependent endocytosis, and to activate β-oxidation and a switch from glycolysis in M1 macrophages to oxidative phosphorylation (oxphosph) in M2 macrophages (center) (highlighted by a green block); (6) participation in selective autophagy initiated from the endoplasmic reticulum (ER) and linked to the mitochondrial membrane via the PI3P site (shown for de novo synthesized ALP in the enterocyte, top left) (highlighted by a blue block). Notes: NOX1—NADPH-dependent oxidase induces ROS production in non-phagocytic cells (enterocytes); MAPK-NF-kB-induced pathway for expression of MMP9-like metalloproteases and nitric oxide synthase iNOS, which damage tight junction proteins between enterocytes and are responsible for hyperpermeability; NOX2—NADPH-dependent oxidase induces ROS production and NF-kB-mediated pathways for expression of pro-inflammatory proteins in phagocytic immune cells; PTEN (phosphatase and tensin homolog)—a dual lipid phosphatase, antagonizes the PI3K-Akt-mTOR pathway, mitochondrial function, and pro-inflammatory innate immune response through its lipid phosphatase activity after allosteric dephosphorylation of four C-terminal phosphate groups. The ALP pathways are classified into four categories, distinguished by their putative mechanism of action, and represented by colored blocks according to the Chapters: “Alkaline Phosphatase Inhibit LPS-TLR4 Binding and TLR4-mediated NF-kB Signaling” (yellow block); “Alkaline Phosphatase Modulate Purinergic Signaling” (rose block); “Alkaline Phosphatase Modulate Pro-Inflammatory Metabolism” (green block); “Alkaline Phosphatase-Phosphate Complexes Trigger Autophagy and Endocytosis” (blue blocks).

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
