# Peer review of "Insights into Alkaline Phosphatase Anti-Inflammatory Mechanisms"

_biomedicines, 2024, doi:10.3390/biomedicines12112502_

Round 1
Reviewer 1 Report
Comments and Suggestions for Authors
This manuscript outlines the role of alkaline phosphatase (ALP), both as an endogenous enzyme and when administered externally, in reducing inflammation across several experimental models of diseases like colitis, liver failure, and ischemia-reperfusion injury. ALP's anti-inflammatory mechanisms include neutralizing lipopolysaccharide (LPS), modulating purinergic signaling, and activating pathways like CD36-mediated β-oxidation, caveolin-dependent endocytosis, and autophagy. These processes contribute to reducing oxidative stress and inflammation.
The quality of this manuscript is very good, although I suggest a little revision for Figure 1. Since there are 4 major sections to introduce the mechanism of ALP's anti-inflammatory activity, is that possible to label them in the figure with different colors of boxes? It could be easier for reader to locate what is discussing. Also, could authors also label (1)-(6) pathways (descried m the legend) in the figure 1?
Author Response
Comments:
This manuscript outlines the role of alkaline phosphatase (ALP), both as an endogenous enzyme and when administered externally, in reducing inflammation across several experimental models of diseases like colitis, liver failure, and ischemia-reperfusion injury. ALP's anti-inflammatory mechanisms include neutralizing lipopolysaccharide (LPS), modulating purinergic signaling, and activating pathways like CD36-mediated β-oxidation, caveolin-dependent endocytosis, and autophagy. These processes contribute to reducing oxidative stress and inflammation.
The quality of this manuscript is very good, although I suggest a little revision for Figure 1. Since there are 4 major sections to introduce the mechanism of ALP's anti-inflammatory activity, is that possible to label them in the figure with different colors of boxes? It could be easier for reader to locate what is discussing. Also, could authors also label (1)-(6) pathways (descried m the legend) in the figure 1?
Response:
I would like to express my gratitude to the esteemed Reviewers for dedicating their time and offering a favorable assessment of the manuscript.
In accordance with the recommendations of Reviewer, Figure 1 has been revised. The pathways described in the legend to Figure 1 have been numbered, and the processes corresponding to the information in the four Chapters have been highlighted in four colors.
Reviewer 2 Report
Comments and Suggestions for Authors
This manuscript aims to examine the anti-inflammatory mechanisms of alkaline phosphatase. The majority of the articles discussed in this review have been referenced to clarify the process of inflammation and the immune response pathway that includes alkaline phosphatase. The possible anti-inflammatory actions of nonspecific phosphomonoesterase, as well as cellular or extracellular alkaline phosphatase, on innate immune and epithelial cells, have been explained, and Figure 1 demonstrates these mechanisms. The review thoroughly discusses all aspects of the anti-inflammatory mechanisms of alkaline phosphatase. This review will be a crucial point of reference for immunologists and practitioners. However, a few slight adjustments need to be made.
1. Avoid giving a conclusion in the abstract since it is already provided at the end of the article.
2. Abbreviation should be given in full and abbreviation in the bracket where it is given first, and then only abbreviation should be used throughout the manuscript (see line 14 and line 20 for alkaline phosphatase (ALP). Be consistent throughout the manuscript.
3. Avoid using only abbreviations in the abstract (e.g., in line 21: LPS-induced)
4. Figure 1 is labeled as sectioned (1) and (2) in the legend but not shown in the figure itself.
5. There are some typos (Line 413: py-rimidin, line 419: en-dogenous, line 210; nti-inflammatory)
6. Some minor English mistakes need to be corrected.
Author Response
Comments:
This manuscript aims to examine the anti-inflammatory mechanisms of alkaline phosphatase. The majority of the articles discussed in this review have been referenced to clarify the process of inflammation and the immune response pathway that includes alkaline phosphatase. The possible anti-inflammatory actions of nonspecific phosphomonoesterase, as well as cellular or extracellular alkaline phosphatase, on innate immune and epithelial cells, have been explained, and Figure 1 demonstrates these mechanisms. The review thoroughly discusses all aspects of the anti-inflammatory mechanisms of alkaline phosphatase. This review will be a crucial point of reference for immunologists and practitioners. However, a few slight adjustments need to be made.
- Avoid giving a conclusion in the abstract since it is already provided at the end of the article.
- Abbreviation should be given in full and abbreviation in the bracket where it is given first, and then only abbreviation should be used throughout the manuscript (see line 14 and line 20 for alkaline phosphatase (ALP). Be consistent throughout the manuscript.
- Avoid using only abbreviations in the abstract (e.g., in line 21: LPS-induced)
- Figure 1 is labeled as sectioned (1) and (2) in the legend but not shown in the figure itself.
- There are some typos (Line 413:py-rimidin, line 419: en-dogenous, line 210; nti-inflammatory)
- Some minor English mistakes need to be corrected.
Response:
I would like to express my gratitude for your time and for your positive assessment of the manuscript.
Comments 1:
Avoid giving a conclusion in the abstract since it is already provided at the end of the article
Response 1:
I am grateful for your assistance. The abstract was formatted in accordance with the specifications set out in the manuscript template, which included the aforementioned heading.
However, in line with the Reviewer's recommendation, it has been removed.
Comments 2 and 3:
Abbreviation should be given in full and abbreviation in the bracket where it is given first, and then only abbreviation should be used throughout the manuscript (see line 14 and line 20 for alkaline phosphatase (ALP). Be consistent throughout the manuscript.
Responses 2 and 3:
I am grateful to you for bringing this to our attention. The abbreviations used in the abstract and throughout the text have been checked and spelled out in full for the first time they are mentioned.
Comments 4:
Figure 1 is labeled as sectioned (1) and (2) in the legend but not shown in the figure itself.
Response 4:
In line with the Reviewer's suggestion, all section numbers referenced in the legend have been included in the figure (1-6).
Comments 5:
There are some typos (Line 413:py-rimidin, line 419: endogenous, line 210; nti-inflammatory)
Response 5:
I appreciate your attention. Please be advised that the typos have been corrected.
Comments 6:
Some minor English mistakes need to be corrected.
Response 6:
Please accept our sincerest thanks. Any mistakes in the English text have been corrected.